# Fear of Death during COVID-19 Does Not Explain Post-Infection Depression Symptoms beyond Reported Symptoms during the Infection in COVID-19 Survivors

**DOI:** 10.3390/ijerph192113773

**Published:** 2022-10-23

**Authors:** Leoni-Johanna Speichert, Adam Schweda, Oliver Witzke, Margarethe Konik, Hana Rohn, Mark Stettner, Venja Musche, Klaas Herchert, Madeleine Fink, Sheila Geiger, Alexander Bäuerle, Eva-Maria Skoda, Martin Teufel, Hannah Dinse

**Affiliations:** 1Clinic for Psychosomatic Medicine and Psychotherapy, University of Duisburg-Essen, LVR-University Hospital, 45147 Essen, Germany; 2Center for Translational Neuro- and Behavioral Sciences (C-TNBS), University of Duisburg-Essen, 45147 Essen, Germany; 3Department of Infectious Diseases, West German Centre of Infectious Diseases, University of Duisburg-Essen, Essen University Hospital, 45147 Essen, Germany; 4Department of Neurology, University Medicine Essen, University of Duisburg-Essen, 45147 Essen, Germany

**Keywords:** COVID-19 survivors, depression, fear of death, generalized anxiety, mental health, post-COVID-19, long COVID, corona, psychological burden, psychosomatic

## Abstract

The COVID-19 pandemic poses an unprecedented global burden to the general population and, in particular, to individuals who have been infected with SARS-CoV-2. In the context of the discussion about “post COVID-19”, the aim of the study was to advance research on mental health and long-term consequences after COVID-19. In total, 214 COVID-19 survivors (female: 54.2%; hospitalized: 36.7%) participated in the repeated cross-sectional assessment. In addition to demographic data, mental and somatic symptoms, fear of death at the time of infection, and depressive (PHQ-8) and generalized anxiety symptoms (GAD-7) were assessed. Results showed an increased prevalence of depressive symptoms and symptoms of generalized anxiety compared to observations in the general population prior to the COVID-19 pandemic. Psychological symptoms of depression and reported levels of fear of death during the SARS-CoV-2 infection showed a negative association with the time interval since COVID-19 diagnosis. Furthermore, although fear of death during the acute COVID-19 was related to depression and generalized anxiety, this association was predominantly explained by the presence of mental and somatic symptoms. In conclusion, initial fear of death does not impact mental health beyond the overall symptom burden. Furthermore, depressive symptoms appear to vanish across time since infection.

## 1. Introduction

The first cases of the novel coronavirus called severe acute respiratory syndrome coronavirus 2 (SARS-CoV-2) occurred in China in December 2019 [1]. The number of infections increased rapidly in many countries; as a result, most of the world found itself quickly in a state of emergency. Since the beginning of the pandemic, borders have been closed several times, restrictions have been announced, and life took place almost entirely at home in order to prevent the virus’ spread [2]. Despite these efforts, the pandemic has currently (through 18 October 2022) resulted in 622,389,418 confirmed cases and 6,548,492 deaths worldwide and continues to challenge global health systems [3].

The coronavirus disease (COVID-19) describes the clinical syndrome caused by the SARS-CoV-2 infection. Although the majority of cases are asymptomatic or mild, a large proportion of infected individuals requires treatment in hospitals or even intensive care units [4,5]. However, the pandemic poses more than just physical dangers. Due to the associated restrictions and potential uncertainties, it also has a lasting negative impact on the mental health of the population, which is mostly small but significant [6,7,8,9,10,11].

In addition to the secondary social and public health implications for the general population, it is important to consider COVID-19 sufferers and survivors who are exposed to additional stressors and thus represent a special risk group; the acute infection, with even severe and life-threatening symptoms, may cause a severe and sustained burden. Long-term psychological consequences after acute infections are already known from previous coronavirus outbreaks [12,13]. Therefore, the psychological burden in COVID-19 patients was expected at the beginning of the pandemic, during the pandemic, but also after the acute stage of the disease [13,14]. Uncertainty, fear of the infection’s consequences, stigmatization, and concerns about having infected another person, social isolation, or the unfamiliar situation during the inpatient stay are conceivable triggers for psychological burden after COVID-19 [15,16].

Early on, it was recognized that many patients after COVID-19 continue to suffer from physical and psychological stress even after the acute stage of the disease [17]. Research continues to assess patients’ impairments during and after COVID-19. Increasingly, the term “Post or Long COVID” has emerged, describing symptoms weeks and months after the acute stage of the disease [18]. Due to the variety and novelty of symptoms experienced by SARS-CoV-2-infected individuals long after their infection, there has long been no precise definition [19]. The World Health Organization (WHO) recommends the term “post COVID-19 condition” for the health condition of individuals who have symptoms resulting from probable or confirmed SARS-CoV-2 infection three months after the onset of COVID-19. These symptoms cannot be explained by any other diagnosis and last for at least two months [20].

Due to the increasing number of people suffering from various symptoms after COVID-19, a precise examination of the manifold symptomology is of great importance. This includes the highly relevant assessment of mental health in COVID-19 survivors. Prior studies confirmed increased psychological distress following a SARS-CoV-2 infection [21,22,23,24,25]. In a systematic review and meta-analysis, it was shown that insomnia, fear, and PTSD, followed by depression, somatization, and anxiety, were the most common psychological distresses [26]. Nevertheless, accurate assessment of the psychological burden in individuals after SARS-CoV-2 infection is far from complete.

The primary aim of the present study is to assess the psychological burden of COVID-19 survivors and its dynamics across time, as well as the influence of symptom severity and fear of death during COVID-19 on current levels of burden. To assess psychological burden after COVID-19, symptoms of depression and generalized anxiety were examined.

## 2. Materials and Methods

### 2.1. Procedure

Data were collected by repeated cross-sectional assessments in an outpatient clinic for COVID-19-recovered patients. The interdisciplinary outpatient clinic was established in May 2020 by the Department of Infectious Diseases of the University Hospital Essen in collaboration with the Clinic for Psychosomatic Medicine and Psychotherapy (Essen, Germany). It represents one of the largest outpatient clinics for COVID-19 survivors in Germany and provides structured follow-up care for patients whose symptoms appear to be related to their past infection with SARS-CoV-2. At the appointment, apart from the evaluation of somatic issues, the follow-up included a structured psychosomatic assessment.

### 2.2. Participants

The study included 214 participants who attended the outpatient clinic between 29 April 2020 and 5 March 2021 and completed the psychosomatic assessment. Participants differed in terms of treatment for their acute COVID-19; some did not require medical treatment, while others were treated as outpatients or required hospitalization and treatment in an intensive care unit (see Table 1). Awareness of the outpatient clinic was raised through direct referrals from other physicians/general practitioners, print media, online channels, websites, or a letter of appeal from the local health department.

Inclusion criteria for the study were a COVID-19 diagnosis at least 4 weeks prior, reaching the age of majority (≥ 18 years), and sufficient proficiency in German to complete the assessment. Beyond that, there were no further specific admission or exclusion criteria.

Informed consent was obtained from all participants involved in the study. The study was conducted in accordance to the guidelines of the Declaration of Helsinki and approved by the local Ethics Committees of the University Hospital Essen (20-9307-BO).

### 2.3. Measures

The socio-demographic data included gender, age, marital status, education level, as well as federal state and community size (large city, medium city, small city, rural community). Other questions were related to the SARS-CoV-2 infection; collected information included date of diagnosis and mental and somatic symptoms at that time, duration of prescribed home quarantine, and fear of death during the acute infection (scale 0–100; 0 = no fear at all, 100 = very intense fear). Further questions concerned the type of treatment (no medical treatment, outpatient, inpatient, intensive care) and the measures taken during treatment (medical treatment, isolation, artificial ventilation, and psychological care). The paper-and-pencil questionnaire took about 15–20 min to complete. The questionnaire was given to the patients before the examination in the outpatient clinic for COVID-19 recovered patients and answered by them in the waiting area. After completion, the questionnaire was collected by the examining physician.

Validated measures were used to assess psychological burden at the time of presentation to the outpatient clinic: the Patient Health Questionnaire-8 (PHQ-8) and the Generalized Anxiety Disorder Scale-7 (GAD-7) [27,28,29]. The PHQ-8 consists of eight items assessing the frequency of depressive symptoms in the past two weeks on a 4-point Likert scale (0 = never experiencing the symptom, 3 = experiencing the symptom nearly every day). A total sum score of ≥5, ≥10, ≥15, and ≥20 points indicates a mild, moderate, intermediate, and severe depression symptomatology, respectively [27].

The GAD-7 consists of seven items representing the frequency of anxiety symptoms in the past two weeks on a 4-point Likert scale (0 = never to 3 = almost every day). Sum scores of ≥5, ≥10, and ≥15 indicate mild, moderate, and severe generalized anxiety symptoms, respectively [28,29].

### 2.4. Data Analyses

In order to first generate a demographic and disease historic description of the sample, simple summary statistics were reported. Only complete cases were used for the respective analyses. Furthermore, it was examined whether time after acute infection was associated with the current psychological symptom severity. For this purpose, simple correlation analyses were applied first. Since none of the variables were normally distributed, Spearman correlation coefficients were computed. To assess the robustness of these tests, further robust regression models were computed that included sociodemographic data as covariates. Finally, the association between the symptomatic burden—and particularly the fear of death—during the acute SARS-CoV-2 infection and the mental status of the patients was explored. Here, the amount of symptoms from the mental (e.g., anxiety, feelings of depression, listlessness) and somatic domain (e.g., dyspnea, headache, cough) were summed up and regressed on depression and anxiety scores during the acute COVID-19 infection. Retrospectively assessed fear of death during the SARS-CoV-2 infection was also included in the regression equation, as were time since diagnosis, age, gender, and education as potentially confounding covariates. All data analyses were performed using R (4.0.3, R Core Team, 2021, R Foundation for Statistical Computing, Vienna, Austria) [30]. Robust regression models were computed using the *robustbase* package [31]. The level of significance was set at α = 0.05 (two-sided tests).

## 3. Results

### 3.1. Participants Characteristics

A total of 214 patients participated in the psychosomatic assessment. Among the study participants, 76 (36.7%) patients were hospitalized during the acute stage of COVID-19. All relevant sociodemographic, disease historic, and treatment-related data are shown in Table 1. Table 2 presents the severity of fear of death during COVID-19 as well as the prevalence of depressive symptoms (PHQ-8) and symptoms of generalized anxiety (GAD-7). In total, 205 (95.8%) of the study participants completed the question regarding “Fear of death during COVID-19”. In addition, 181 (84.6%) participants completed the PHQ-8 questionnaire, and 190 (88.79%) participants finished the GAD-7 questionnaire.

### 3.2. Symptoms of Depression and Generalized Anxiety in COVID-19 Survivors after the Acute Infection and Retrospectively Reported Concerns about Death during the Acute Infection

Retrospectively reported concerns about death (rs = −0.224 *p* = 0.002, see Figure 1A) as well as depression scores (rs = −0.173, *p* = 0.026, Cronbach’s α = 0.86, MacDonald’s ω = 0.86, see Figure 1B) decreased with time—a relationship that also remained stable after conditioning with age, gender, and education in a robust regression model. Yet, no such relationship with time elapsed since COVID-19 diagnosis was observed for generalized anxiety (rs = −0.48, *p* = 0.539, Cronbach’s α = 0.9, MacDonald’s ω = 0.91).

### 3.3. Predicting Current Mental Health Status Using Retrospectively Reported Symptoms during SARS-CoV-2 Infection

Next, it was attempted to predict current mental health states using retrospectively reported peri-infection criteria. To this end, the overall amount of mental symptoms (e.g., anger, feelings of depression and listlessness), somatic symptoms (e.g., cough, fever, headache), the time since diagnosis, fear of death during acute COVID-19, along with covariates that condition on sociodemographic criteria were regressed on depression (PHQ-8) and generalized anxiety (GAD-7) scores at the time of presentation to the outpatient clinic. For depression, time since COVID-19 diagnosis was again shown to negatively predict PHQ-8 scores (b = −0.009, t = −2.182, *p* = 0.031). Furthermore, the total sum of mental health-related symptoms experienced during the infection was identified as a strong predictor (b = 0.772, t = 3.356, *p* = 0.001). Similarly, the sum of somatic symptoms was revealed to be a significant but comparatively less strong predictor (b = 0.464, t = 2.063, *p* = 0.041). Importantly, fear of death during COVID-19 infection made no predictive contribution in this model (b = −0.0005, t = −0.28, *p* = 0.978, total model R^2^ = 0.301), despite the fact that it was positively associated with depression scores in a model that only contains fear of death, gender, age, and education (b = 0.049, t = 3.557, *p* < 0.001). Further inspection revealed that this association was no longer significant after conditioning on the sum of somatic (b = 0.024, t = 1.736, *p* = 0.085) or mental symptoms (b ≤ 0.001, t = 0, *p* = 0.999), indicating that there was no effect of fear of death during the acute COVID-19 period that acted beyond the reported number of symptoms that occurred during the SARS-CoV-2 infection.

A very similar pattern was found for generalized anxiety; a significant association between fear of death during acute SARS-CoV-2 infection and higher GAD-7 scores occurred when only gender, age, and education were included as covariates (b = 0.052, t = 2.932, *p* = 0.004). Yet, adding the sum of mental health-related symptoms during acute infection completely abolished this association (b = 0.025, t = 1.361, *p* = 0.175). When only physical symptoms were considered, the association remained unchanged (b = 0.047, t = 2.987, *p* = 0.003). In addition, when both were considered in the same model, fear of death was not significant (b = 0.025, t = 1.393, *p* = 0.166, total R^2^ = 0.307).

## 4. Discussion

To our knowledge, this is the first study in Germany to examine the course of long-term psychological consequences in COVID-19 survivors. Regardless of the course of COVID-19, the mental and somatic symptoms, and the experienced fear of death during the acute infection, significant psychological burden can be observed for weeks after the onset of the disease. This burden is manifested by symptoms of depression and generalized anxiety. The interval between COVID-19 and the examination date varied among participants, allowing observation over a longer period of time. In this study, it was shown that individuals with a longer time interval to their COVID-19 diagnosis had, on average, lower scores for depressive symptoms and for experienced fear of death during COVID-19 than the rest of the cohort. Furthermore, the number of mental and physical symptoms at the time of acute SARS-CoV-2 infection predicted long-term psychological burden in COVID-19 survivors. However, in this context, fear of dying from COVID-19 could not explain the depressive symptoms in COVID-19 survivors.

Several studies have already indicated negative effects of the pandemic on the mental health of the general population [6,7,8,9,10,11,32]. However, COVID-19 survivors are at even greater risk because of their prior disease. Hospitalized patients, especially those treated in an intensive care unit and even artificially ventilated due to the severity of their disease, appear to be at particularly high risk [24]. To date, a large and growing proportion of the world’s population has already been infected with SARS-CoV-2 [3]. After the acute stage of the disease, a proportion of infected individuals suffers long-term consequences that can manifest not only in physical and cognitive burdens but also, as demonstrated in the present study, in a psychological burden [21,22,23,24,25,26]. Due to the various, fluctuating, and apparently often long-lasting symptoms, the exact extent is still unknown.

Based on this negative association between time interval after COVID-19 and depressive symptoms or fear of death, it could be hypothesized that COVID-19 survivors are able to cope with their burden over time—resulting in a reduction of the psychological burden. A previous study with the same cohort has already shown that COVID-19 survivors, particularly those with COVID-19-related trauma symptoms, showed an improvement in their sense of coherence over time after infection [33]. Nevertheless, the symptom scores of the psychological burden after COVID-19 are still above those of the general population. Generally, depressive symptoms and symptoms of generalized anxiety appear to be more prevalent in the infectiological sample [27,34]. While 8.58% of the German population exceeded the cutoff of 10 on the PHQ-8 in a 2009 study, 34.80% of the patient group reached and exceeded this threshold [27]. In 2017, 5.90% of the German general population reached the cut-off criterion of 10 on the GAD-7 [34]. In the present study, such a cut-off was detected in 22.11% of COVID-19 survivors.

Derived from the results, the question “What caused the psychological burden?” arises. Thus, it could be assumed that severe anxiety and severe mental or somatic symptoms during COVID-19 would increase the psychological burden post COVID-19 and lead to psychological symptoms.

In line with general expectations, the number of mental and physical symptoms at the time of acute SARS-CoV-2 infection proved to be a predictor of psychological burden weeks after COVID-19. In addition, fear of death at the time of COVID-19 infection was not a significant predictor after conditioning on the sum of physical or mental symptoms, suggesting that fear of death during the acute stage of COVID-19 has no impact beyond the reported number of symptoms experienced during SARS-CoV-2 infection.

Due to the lack of scientific evidence to date, the causes of psychological burden in COVID-19 survivors can only be conjectured. One approach could be that patients with COVID-19 usually have to cope with the infection in isolation. In addition to the stress and loneliness caused by isolation, COVID-19 symptoms can be perceived as very traumatic and lead to the development of worries and anxiety. Another possible explanation is that patients are constantly reminded of COVID-19 during the ongoing pandemic, e.g., through the media, government-imposed restrictions, in private settings, or even through persistent somatic symptoms [35]. In light of this, increased consumption of COVID-19 information may lead to increased negative evaluation of the patient’s past COVID-19. In retrospect, the disease is perceived as more burdensome and distressing, leading to increased psychological burden after COVID-19.

Regarding the diversity of the described symptoms in the context of post COVID-19, it should be discussed whether a possible explanation between the increased psychological burden and the intense somatic symptoms described without a clear medical correlate could be based on a bodily distress syndrome [36]. A bodily distress syndrome describes the presence of somatic symptoms that are perceived as highly distressing, receive an excessive and unwarranted high-level of attention, and lead to increased health care utilization [37,38]. Anticipation of long-term consequences leads to increased interoceptive awareness, resulting in the perception of presumed symptoms and their attribution to post COVID-19. Therefore, a critical approach to preventing the development of a bodily distress syndrome in COVID-19 patients is to educate the general population about the extremely low likelihood of developing long-term physical consequences. In addition, the training of medical personnel on this issue is of great importance, as is close interdisciplinary cooperation between the psychological and physical departments of medicine.

With this in mind, the research findings presented highlight the need for standardized referral of COVID-19 survivors to psychological counselling services. Low-threshold interventions should be quickly established to provide support and assistance to patients. Digital online intervention programs, exchange platforms, and online consultations already represented a major benefit during the pandemic [39,40,41,42,43]. In addition, our results raise questions that should be answered in further research, for example, whether people with increased psychological burden after COVID-19 also show more somatic symptoms. Furthermore, it would be important to investigate whether patients who showed mental distress after COVID-19 also showed increased mental pre-existing conditions before the infection with SARS-CoV-2.

The present study provides new, significant, and much-needed insights into the mental health and its course over time in COVID-19 survivors. However, limitations must be considered. First, it should be mentioned that the data collection is based on repeated cross-sectional assessments, which does not allow conclusions to be made about individual changes in psychological stress. By using a general questionnaire at only one point in time, symptoms and feelings at the time of a COVID-19 diagnosis could only be asked by recollection, which may bias the data. In addition, it was not recorded how many patients in the post-COVID-19 outpatient clinic declined to be interviewed. Furthermore, it should be noted that participants presented to the aftercare outpatient clinic on the advice of their physician or on their own initiative. It can be assumed that these patients are more likely to have mental or somatic symptoms than patients who did not visit the outpatient clinic.

## 5. Conclusions

An encouraging conclusion is that depressive symptom scores were lower in individuals whose infection with SARS-CoV-2 occurred further back in time than in individuals whose infection occurred only a few weeks prior to the study. This finding indicates that patients are able to recover from the effects of COVID-19 and to cope with the disease and its consequences.

Nevertheless, attention must be paid to COVID-19 survivors in follow-up care. In particular, patients who have experienced severe physical but also mental stress during their acute illness should be offered psychological aftercare in close interdisciplinary cooperation with somatic disciplines.

## Figures and Tables

**Figure 1 ijerph-19-13773-f001:**
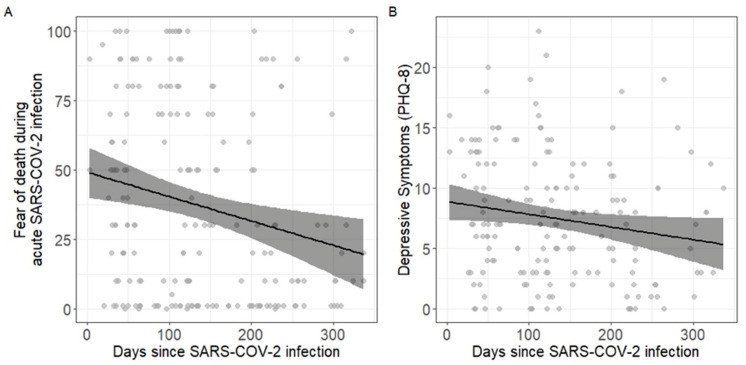
(**A**) Association between the fear of death during the acute SARS-CoV-2 infection (0 = not at all, 100 = very high) and days passed since diagnosis of COVID-19; (**B**) association between depressive symptoms (sum score PHQ-8) with days passed since diagnosis of COVID-19.

**Table 1 ijerph-19-13773-t001:** Demographics. Values are presented as n (percent).

		Overall n = 214 n (%)
Age Categories	18–24 years	9 (4.3)
	25–34 years	21 (10.1)
	35–44 years	43 (20.8)
	45–54 years	59 (28.5)
	55–64 years	47 (22.7)
	65–74 years	19 (9.2)
	75–84 years	9 (4.3)
Gender	Female	115 (54.2)
Education	University/College	76 (37.3)
	High School Degree	50 (24.5)
	Secondary School Certificate (Realschule)	38 (18.6)
	Secondary School Certificate (Hauptschule)	36 (17.6)
	No Degree	2 (1.0)
	Other	2 (1.0)
Quarantined during/after infection		188 (92.6)
Treatment Modality	Hospitalized	76 (36.7)
	Intensive Care Unit Treatment	26 (12.9)
	Underwent Artificial Ventilation	11 (5.8)

**Table 2 ijerph-19-13773-t002:** The severity of retrospectively assessed fear of death during COVID-19, as well as the prevalence of depressive symptoms (PHQ-8) and symptoms of generalized anxiety (GAD-7) at the time of presentation to the outpatient clinic. Values are presented as n (percent) or means ± SD.

Variable	Level	Overall n = 214 n (%)/M ± SD
Fear of death during COVID-19 (mean)		37.22 ± 33.54
Depressive Symptoms: PHQ-8 Score (mean)		7.44 ± 5.52
PHQ-8: Categories: Symptoms of Depression (categories)	None	64 (35.4)
	Mild	54 (29.8)
	Moderate	42 (23.2)
	Moderate to Serve	17 (9.4)
	Serve	4 (2.2)
Symptoms of Generalized Anxiety: GAD-7 Score (mean)		5.36 ± 4.99
GAD-7: Categories: Symptoms of Generalized Anxiety (categories)	None	107 (56.3)
	Mild	41 (21.6)
	Moderate	28 (14.7)
	Serve	14 (7.4)

## Data Availability

The raw data supporting the conclusions of this article will be made available by the authors on request.

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
