# Peer review of "Fear of Death during COVID-19 Does Not Explain Post-Infection Depression Symptoms beyond Reported Symptoms during the Infection in COVID-19 Survivors"

_ijerph, 2022, doi:10.3390/ijerph192113773_

Round 1

Reviewer 1 Report

Much more detail is needed on the delivery and data collection procedure: when is the measurement done?how is the data collected: are the patients given the measurement instruments and then collected?did the patients fill it out in front of any researcher? how were the 214 participants chosen? What was the response rate? Etc.

If you ask them about the fear of dying they had at the time they had the disease, after 300 days (data you get from the figures, since they don't indicate it anywhere else), the bias is huge because you have to remember the feelings you had a long time ago, how do you solve that bias?

Perhaps it would be interesting to compare the data according to the severity of the symptoms, in an objective manner, not only with perceptions and opinions of the users interviewed. It is even very useful to know their state of health after suffering the disease; a plausible hypothesis is that those who have more symptoms or worse physical health after COVID-19 may have greater psychological problems.

Author Response

Dear Reviewer,
Thank you very much for your comments and suggestions. They have been very helpful to us in revising our manuscript. 
Please see the attachment to view our responses. 

Sincerely,

Prof. Dr. Martin Teufel, Ms. Leoni-Johanna Speichert and Dr. Hannah Dinse 

Reviewer 2 Report

Thank you for the opportunity to review this study entitled “Fear of death during COVID-19 does not explain post-infection depression symptoms beyond reported symptoms during the infection in COVID-19 survivors” (ijerph-1971729).

The study focused on the psychological effect of the COVID-19 pandemic, particularly focusing on long-term consequences after COVID-19. A sample of 214 COVID-19 survivors was involved in the research.

In my opinion, the research topic is relevant, and the study is interesting. Parallelly, there are some issues that need to be addressed before the paper will be suitable for publication.

1.     Introduction: In my opinion, it would be good to refer to trend or longitudinal studies, if any. Since the authors frame this study considering the impact that COVID-19 has on a psychological level, I suggest some research to propose a comprehensive framework in the introduction, which should be supplemented with further literature search by the authors:

- Hyland et al., 2021; doi: 10.1016/j.psychres.2021.113905.

- Gori & Topino, 2021; doi: 10.3390/ijerph18115651

- Wang et al., 2020; doi: 10.1016/j.bbi.2020.04.028

To find the suggested articles, the authors can use this source: https://www.doi.org/

2.     Information on the internal consistency (e.g., alpha) for each measure in the sample used in this research should be provided.

3.     The authors stated: “Only complete cases were used for the respective analyses”. What is the percentage of complete cases?

4.     Please, provide suggestions for future research based on these results

5.     Please, don’t start a new section (e.g., "Conclusions" section) with “However”.

In general, I really enjoyed this paper, which seems to be well-structured, interesting, and pleasant to read. In my opinion, after the authors make small changes, it will be ready to be published.

Best wishes

Author Response

(The authors gave the same response as above.)
